

# Contrasting solubility and speciation of metal ions in total suspended particulate matter and fog from the coast of Namibia

Chiara Giorio[1,2]*, Anne Monod[3], Valerio Di Marco[2], Pierre Herckes[4], Denise Napolitano[4], Amy Sullivan[5], Gautier Landrot[6], Daniel Warnes[1], Marika Nasti[1], Sara D'Aronco[1], Agathe Gérardin[3], Nicolas Brun[3], Karine Desboeufs[7], Sylvain Triquet[7], Servanne Chevaillier[8], Claudia Di Biagio[7], Francesco Battaglia[7], Frédéric Burnet[9], Stuart J. Piketh[10], Andreas Namwoonde[11], Jean-François Doussin[8], and Paola Formenti[7]

[1] Yusuf Hamied Department of Chemistry, University of Cambridge, Lensfield Road, Cambridge, CB2 1EW, United Kingdom
[2] Dipartimento di Scienze Chimiche, Università degli Studi di Padova, via Marzolo 1, 35131 Padova, Italy
[3] Aix Marseille Univ, CNRS, LCE, Marseille, France
[4] School of Molecular Sciences, Arizona State University, Tempe, Arizona 85287, USA
[5] Department of Atmospheric Science, Colorado State University, Fort Collins, CO 80523, USA
[6] Synchrotron SOLEIL, L'Orme des Merisiers, Saint-Aubin, France
[7] Université Paris Cité and Université Paris Est Creteil, CNRS, LISA, F−75013 Paris, France
[8] Université Paris Est Creteil and Université Paris Cité, CNRS, LISA, F−94010 Créteil, France
[9] CNRM, Université de Toulouse, Météo-France, CNRS, Toulouse, France
[10] Unit for Environmental Science and Management, North-West University, Potchefstroom, North-West, South Africa
[11] Sam Nujoma Marine and Coastal Resources Research Centre, University of Namibia, Henties Bay, Namibia

*Correspondence to*: Chiara Giorio (chiara.giorio@atm.ch.cam.ac.uk)

**Abstract.** The west coast of southern Africa is a region of particular climate interest and a crossroad for aerosols of different origins as well as fog occurrences. In this study, we present a comparison between the solubility of trace metals in pairs of total suspended particulate (TSP) and fog water samples collected in Henties Bay, Namibia, during the AErosols, Radiation and CLOuds in southern Africa (AEROCLO-sA) field campaign in September 2017. From inductively coupled plasma mass spectrometry measurements, we found that Al, Fe, Ni, Cu, and Cr have enhanced solubility in fog samples compared to the TSP samples. We found that thermodynamic modelling predicts the formation of soluble complexes with inorganic and organic ligands in fog for Cu, Cr, and Ni, but it would predict Al and Fe to precipitate as hydroxides given the neutral pH of fog. Contrastingly, X-ray absorption near edge structure measurements showed the presence of oxalate of Fe complexes that could explain its enhanced solubility in fog samples, despite a neutral pH. In addition, transmission electron microscopy and dynamic light scattering measurements revealed the presence of nano-sized colloidal particles containing Fe and Al in filtered fog samples that may appear as soluble in ICP-MS measurements. We hypothesise that those complexes are formed in the early stages of particle activation into droplets when water content and, therefore, pH are expected to be lower and then remain in fog in a kinetically stable form or lead to the formation of colloidal particles.



## 1 Introduction

Aerosols contribute the largest uncertainty to global radiative forcing budget estimates (IPCC, 2021) due to the lack of information related to global aerosol distributions, composition, and ageing effects in the atmosphere, all of which affect aerosol radiative properties. Natural aerosols from aeolian erosion of arid and semi-arid continental surfaces, marine emissions, and forest fires constitute an essential piece of the puzzle, without which our ability to predict Earth's climate evolution over time diminishes and the development of adaptation strategies for future climate change remains limited

(Bauer et al., 2022).

The west coast of southern Africa is a region of particular climate interest however yet understudied. It is the crossroad for aerosols of different origins and chemical compositions, including mineral dust emitted predominantly from the coastal riverbeds and the gravel plains in Namibia (Dansie et al., 2018; von Holdt and Eckardt, 2018; Shikwambana and Kganyago, 2022; Vickery and Eckardt, 2013), sea spray and marine biogenic emissions from the Benguela upwelling system (Giorio et

al., 2022a; Klopper et al., 2020), ship traffic emissions and dust from mining (Klopper et al., 2020; Tournadre, 2014), and long-range transported biomass burning aerosols from central Africa (Flamant et al., 2022; Formenti et al., 2018, 2019; Gupta et al., 2020; Hecobian et al., 2011; Redemann et al., 2021). It is also an area of the globe where the future climate projections predict severe warming and drought (Engelbrecht et al., 2024; Trisos et al., 2023), affecting the most sensitive regional environmental features, such as the oceanic Benguela upwelling system and stratocumulus cloud deck (Jarre et al.,

2015; Louw et al., 2016; Tyson and Preston-Whyte, 2000), but also the coastal fog resulting from the advection of low-stratus and moist cold air masses from the ocean (Andersen et al., 2020), and the vector of water to the hyper-arid Namibian ecosystems (Gottlieb et al., 2019; Henschel et al., 2019).

The west coast of southern Africa appears henceforth as an ideal natural laboratory for investigating the composition of aerosols and their metal chemical speciation, i.e., the chemical form by which metals, and in particular iron, are present in

the aerosol and dissolved in a solution, and the way by which they condition their ability to form fog and cloud droplets.

In this study, we take advantage of the first simultaneous measurements of aerosol and fog composition conducted at a ground-based site in Henties Bay, Namibia, during the AErosols, Radiation and CLOuds in southern Africa (AEROCLO-sA) field campaign in August-September 2017 (Formenti et al., 2019). In particular, we focus on the period of 3-11 September which was characterised by the occurrence of several fog events as well as by a synoptic-scale circulation that brought

northeasterly winds and continental air masses to the site, thus transporting mining dust from northern Namibia and possibly farther north from the Zambian Copper Belt (Formenti et al., this issue). A previous study in this area found that the fractional solubility of iron can reach values of up to 20% due to the photo-reduction processes of marine biogenic emissions (Desboeufs et al., 2024). As a complement to those findings, in this paper, we present a comparison between the solubility of trace metals in pairs of total suspended particulate (TSP) and fog water samples, aiming to understand the role of particulate

matter in fog droplet formation and the chemical processes facilitating the transfer of those aerosol particles to the aqueous phase in fog. In this respect, we explore the formation of metal-ligand complexes (compounds consisting of a central metal





ion coordinated by molecules or ions, the ligands), which could be promoted by the strong oceanic biogenic activities of the Benguela upwelling, including unexpected compounds such as butenes (Giorio et al., 2022a), as well as by fog events. Indeed, many water-soluble organic and inorganic compounds already present in the aerosol have coordinating properties

towards metal ions, and they can increase their solubility, bioavailability, and light-absorption characteristics, as demonstrated by previous studies in urban, rural, and desert environments (Deguillaume et al., 2005, 2010; Desboeufs et al., 2005; Giorio et al., 2017, 2022b; Paris et al., 2011; Paris and Desboeufs, 2013; Tapparo et al., 2020). In addition, it has already been demonstrated that atmospheric processing can lead to increasing solubility of iron in mineral dusts (Buck et al., 2006; Cwiertny et al., 2008; Gao et al., 2019; Longo et al., 2016; Rodríguez et al., 2021; Shi et al., 2012; Winton et al.,

75    2015).

## 2 Experimental methods

The AEROCLO-sA field campaign took place in August-September 2017 at the Sam Nujoma Marine and Coastal Resources Research Centre (SANUMARC) of the University of Namibia at Henties Bay (22°6′S, 14°30′E; 20 m above sea level) (Formenti et al., 2019). The aerosol measurements were conducted by the PortablE Gas and Aerosols Sampling Units

(PEGASUS) mobile facility as detailed in the companion paper by Formenti et al. (this issue). A detailed description of the chemical composition of the TSP samples can be found in Formenti et al. (this issue), while a detailed description of fog microphysics and chemical composition is discussed in section 0.

Four aerosol samples including particles up to 40 µm in aerodynamic diameter (Rajot et al., 2008), hereafter named total suspended particulate (TSP), were collected in parallel during daytime and night-time (approximately from 07:00 to 17:00

UTC and 17:30 to 06:30 UTC, respectively): one on 47-mm diameter Teflon membranes (Zefluor®, 2 µm diameter pore size), two on Nuclepore® polycarbonate filters (one of 37-mm and one of 47-mm diameter, 0.4 µm pore size), and the fourth on 47-mm diameter quartz membranes (Pall, 2500QAT-UP Tissuquartz). Field blanks were also collected (filters mounted on the sampling lines without activating the airflow). Details of preparation and storage, as well as analysis of the elemental composition, water-soluble inorganic and organic species, soluble metals, and organic and elemental carbon are described in

detail in Formenti et al. (this issue).

In the following paragraphs, we describe the strategy of fog and seawater collection and analysis.

The list of aerosol samples and concomitant fog samples relevant to this paper is presented in Table 1.





**Table 1. Dates and times of collection of concurrent aerosol and fog samples at Henties Bay during the AEROCLO-sA campaign in**
**September 2017. Dates are reported as "dd/mm/yy", time is in UTC.**

| Fog Sample ID | H603 | H701 | H801 | H901 | H902 | H1001 | H1002 | H1003 | H1101 |
|---|---|---|---|---|---|---|---|---|---|
| Start Date | 06/09/17 | 07/09/17 | 08/09/17 | 09/09/17 | 09/09/17 | 10/09/17 | 10/09/17 | 10/09/17 | 11/09/17 |
| Start Time | 16:40 | 17:22 | 18:15 | 07:34 | 15:29 | 05:13 | 12:43 | 16:30 | 05:32 |
| End Date | 07/09/17 | 08/09/17 | 09/09/17 | 09/09/17 | 10/09/17 | 10/09/17 | 10/09/17 | 11/09/17 | 11/09/17 |
| End Time | 06:45 | 12:52 | 07:34 | 10:05 | 05:13 | 09:02 | 16:30 | 05:30 | 08:55 |
| TSP Sample ID | S-26 | S-28 | S-31 | S-32 | S-33 | S-34 | | S-35 | S-36 |
| Start Date | 06/09/17 | 07/09/17 | 08/09/17 | 09/09/17 | 09/09/17 | 10/09/17 | | 10/09/17 | 11/09/17 |
| Start Time | 16:10 | 16:22 | 16:30 | 07:24 | 16:38 | 07:05 | | 16:29 | 05:30 |
| End Date | 07/09/17 | 08/09/17 | 09/09/17 | 09/09/17 | 10/09/17 | 10/09/17 | | 11/09/17 | 11/09/17 |
| End Time | 06:29 | 07:23 | 07:11 | 16:21 | 06:51 | 16:18 | | 05:18 | 15:37 |

## 2.1 Fog and seawater sample collection

Fog samples were collected using two pre-cleaned Caltech Active Strand Cloud Water Collectors (CASCC), with a droplet size collection efficiency of 50% at 3.5 µm in diameter (Demoz et al., 1996). One of the collectors was a stainless-steel
version, ss-CASCC, described by Herckes et al. (2002) while the other was a "classic" Teflon® CASCC developed by Demoz et al. (1996). The collectors were run in parallel, at an airflow rate of about  25  $m^3 min^{-1}$, and were thoroughly cleaned before sampling using ultrapure water (>18 MΩ cm, Millipore) for the CASCC, and acetone (HPLC grade 99.9%) then ultrapure water for the ss-CASCC. Blank samples were taken routinely between each fog event, by nebulizing ultrapure water on the CASCC and the ss-CASCC. Samples collected with the ss-CASCC were analysed for organic acids, while the
samples collected with the CASCC were analysed for pH, inorganic ions, and elements. Samples for major ions analysis were prepared immediately after collection from the CASCC collector by pouring 1 mL in a plastic vial and stored at -18 °C. Samples for organic acids analysis were prepared immediately after collection from the ss-CASCC collector by pouring 950 µL sample of non-filtered fog water and 50 µL chloroform in a 1.5 mL glass vial and stored at 4 °C. Aliquots for soluble trace metal analysis were prepared by filtering the samples through a 0.22 µm filter (PES, Millipore) and acidifying them
with high purity nitric acid (EMD Chemicals, 67–70% Omni Trace Ultra) to a pH less than 2.

Nine fog samples were collected (Table 1) through five fog episodes (see section 0 for details of the fog occurrences). In addition to these fog samples, two samples of coastal seawater were collected on the seashore in precleaned HDPE bottles on September 10[th] (at 16:30 UTC) and 11[th] (at 13:17 UTC). They were aliquoted and analysed in the same way as the fog samples.



## 2.2 Fog microphysics and chemical analysis

Cloud droplet measurements were conducted with a fog monitor 100 (FM-100, Droplet Measurement Technologies Inc., Boulder, CO, USA) located on a scaffolding about 2.5 m high. This forward-scattering spectrometer probe provides droplet number concentration, size distribution, and liquid water content for particle sizes with an optical equivalent diameter of 2 to 50 μm. Droplet size is calculated on the basis of laser-scattered light and employing the Mie theory for spherical water particles with a refractive index of 1.33 and is classified into 32 bin sizes of width of 2 or 3 μm. The instrument was calibrated with glass beads of known diameters and refractive index. Despite large uncertainties (Guyot et al., 2015; Spiegel et al., 2012), it is one of the most commonly used instruments to document the microphysical properties of fog ((Gonser et al., 2012; Gultepe et al., 2006, 2009; Mazoyer et al., 2022; Niu et al., 2012; Wagh et al., 2023) among others). Data were recorded every second and subsequently aggregated to 1-minute averages.

To determine the amount of water collected with the CASCC and ss-CASCC, sample bottles were weighed before and after sampling.

The pH was measured by a pH meter (9110DJWP pH Probe from Thermo Scientific) as soon as possible after sampling on non-filtered samples. The pH meter was calibrated daily using pH 4.01, 7.00, and 10.01 buffers.

Water-soluble ions were analysed by ion chromatography (IC, Dionex Corporation). A dual channel Dionex ICS-3000 ion chromatograph equipped with conductivity detectors was used to measure the cations and anions. For the cation analysis, an eluent generator provided a concentration of 20 mM methanesulfonic acid at a flowrate of 0.5 mL/min to perform the separation on a Dionex IonPac CS12A analytical column (3 x150 mm).  The anions analysis was performed on a Dionex IonPac AS14A analytical column (4 x 250 mm) using 1 mM sodium bicarbonate/8 mM sodium carbonate eluent at a flowrate of 1 mL/min. The complete run time for both was 17 min. No fog samples showed concentrations below detection limits, except for phosphate (2 fog samples, all blanks and the sea samples).

Organic acids including acetate, propionate, formate, methanesulphonate (MSA), glutarate, succinate, malonate, maleate, and oxalate were analysed using a Dionex DX-500 IC with a gradient pump, conductivity detector, and self-regenerating anion suppressor.  A 20-fold dilution was used for formate quantification. Separation was performed using a Dionex AS-11HC analytical (4×250 mm) column using a sodium hydroxide gradient at a flow rate of 1.5 mL/min. The complete run time was 65 min.

For soluble trace metal determination, samples were analysed for 40 trace metals using a Thermo Electron X-Series 2 quadrupole inductively coupled plasma mass spectrometer (ICP-MS).

## 2.3 XANES analysis

Iron speciation, including the identification of the presence of iron-organic ligand complexes and the partitioning of iron species in the II- and III-oxidation state, was investigated by XANES at the Fe K-edge. The XANES analysis was performed at the SAMBA (Spectroscopies Applied to Materials based on Absorption) line at the SOLEIL synchrotron facility in Saint-





Aubin, France (Briois et al., 2011). A Si(220) double-crystal monochromator was used to produce a monochromatic X-ray beam, which was 4000 x 1000 $\mu m^2$ in size at the focal point. The energy range was scanned from 7050 eV to 7350 eV at a step size of 0.2 eV.

Standards of Fe(II) and Fe(III) oxides, hydroxides, salts, and oxalate complexes (see Formenti et al., this issue for details) were prepared as round pellets of 10 mm in diameter in a matrix of Boron Nitride (BN, molar mass 24.819 g mol$^{-1}$) using an automatic hydraulic Press (Atlas Power T8, Specac), available at the chemistry laboratory at SOLEIL, and operated at a pressure of 5 Tons. The pellets were analysed in transmission mode. The TSP and fog samples were analysed in fluorescence mode on the same energy range and step resolution as the standard references. As in Formenti et al. (2014) and Caponi et al.

(2017), samples were mounted in an external setup mode. The TSP samples were analysed as detailed in Formenti et al. (this issue). For fog samples, several aliquots of 100 µL solution were deposited on a Kapton sticky film and left evaporating until a significant deposit was observed visually and the minimum Fe concentration of 0.4 µg cm$^{-2}$ was obtained. The deposit was finally protected by sticking another portion of Kapton film on top of it, forming a pellet ready for analysis.

Data analysis was conducted with the FASTOSH software package developed at SOLEIL (Landrot and Fonda, 2024). The

Fe speciation was obtained by the least squares fit of the measured XANES spectra based on the linear combination of reference spectra of oxides, clays, oxalates of iron(II) and (III) and pyrite. Fits were conducted on the first derivative of the normalised spectral absorbance in the energy region between 7100 to 7180 eV, corresponding to -30 and +50 eV of the K-edge. The results of the fits were evaluated on the basis of the resulting chi-square ($\chi^2$). Additional details on reference standards and data analysis are reported in Formenti et al. (this issue).

### 2.4 TEM and DLS measurements of fog samples


The coarse fraction of a fog sample was filtered out by aspirating a fog sample through a 0.2 µm filter. The supernatant was then allowed to sediment, to remove any small but colloidally unstable particulate matter, by resting an Eppendorf tube upright for 24h. The resultant suspension was then sampled for analysis.

A Malvern Panalytical Zetasizer Nano was utilised to perform dynamic light scattering (DLS) on the prepared fog samples.

The angle and intensity of light scattering caused by suspended particles can be correlated to particle size, thereby investigating the hydrodynamic particle size distribution of the suspension and to confirm the presence of a colloidally stable phase.

Fog samples were prepared in the same manner also for Transmission electron microscopy (TEM). A 35 µL droplet of the suspension was deposited onto a TEM grid (Cu-C, 300 mesh), before blotting off the excess with filter paper. A Thermo

Scientific (FEI Company) Talos F200X G2 TEM was used to capture images of the nanoparticles in the fog samples. Energy dispersive X-ray spectroscopy (EDS) was used to identify chemical elements occurring within the aforementioned nanoparticles and to perform elemental mapping within a nanoparticle, to understand the distribution of elements. A Super-X EDS detector system with four windowless silicon-drift detectors was utilized for EDS analysis.



### 2.5 Ancillary measurements

A weather detector (Vaisala PWD 22/52) provided 1 min average horizontal visibility from the forward scatter measurement principle.

Standard meteorological parameters (wind speed and direction, air temperature and relative humidity) were measured by a compact weather station (models CE155N, CE157N, BA711, Cimel Electronique, Paris, France) installed at 10 m above ground level on an extendable mast on the roof of the PEGASUS facility.

## 3 Data analysis

### 3.1 Calculation of pH and liquid water content of aerosols

Average aerosol liquid water (ALW) content and pH for the aerosol fraction were calculated using the comprehensive version of the Extended Aerosol Inorganics Model E-AIM IV (Clegg et al., 1992, 1998; Wexler, 2002). Averaged relative humidity (RH) and temperature over the sampling period and aerosol concentrations of $NH_4^+$, $Na^+$, $Cl^-$, $NO_3^-$, and $SO_4^{2-}$ were

used as input for the calculation. Ion balance was achieved by adding an appropriate amount of $H^+$ or by converting an appropriate amount of $NH_4^+$ into $NH_3$. Acetic acid and oxalic acid were also included in the calculation and allowed to dissociate in solution (Clegg et al., 2003). Partitioning equilibria with gas components and solid precipitation were also considered. $K^+$, $Ca^{2+}$, $Mg^{2+}$, and $PO_4^{3-}$ were not considered in the calculation as these species are not included in E-AIM. Including formic acid in the calculations did not show any difference, therefore it was ignored.

### 3.2 Speciation of metal ions in deliquescent aerosol

Chemical speciation of metal ions in deliquescent aerosol was computed using the model Visual MinteQ 3.1 (available online at https://vminteq.com). For the speciation, the following species were considered: $Na^+$, $NH_4^+$, $K^+$, $Mg^{2+}$, $Ca^{2+}$, $Cl^-$, $NO_3^-$, $SO_4^{2-}$, $PO_4^{3-}$, oxalate, acetate, Al, Cu, Ni, Mn, Cr, Fe, Pb, and Zn. For metal ions, the soluble fraction measured with ICP-MS was used. The oxidation state of the latter metal ions was assumed to be the following: Al(III), Cu(II), Ni(II),

Mn(II), Cr(III), Pb(II), and Zn(II). In the case of Fe, we assumed that Fe(II) and Fe(III) were present in a ratio 1:1 as used in previous studies (Giorio et al., 2022b; Scheinhardt et al., 2013a; Shahpoury et al., 2024a). This 1:1 ratio was used to obtain an average picture of Fe speciation considering the dynamic nature of the $Fe^{2+}/Fe^{3+}$ couple in atmospheric aerosol with expected higher $Fe^{2+}$ concentrations during the day and higher $Fe^{3+}$ concentrations during the night (Mao et al., 2017). For the speciation, liquid phase concentrations of the dissolved complex-forming compounds were calculated using the ALW

obtained from E-AIM IV, assuming initially a complete dissolution. The pH was fixed at the value calculated by E-AIM IV, and the mean temperature over the sampling time was used. The speciation model was run according to the procedure described in previous studies (Giorio et al., 2022b; Scheinhardt et al., 2013b). When specific chemical species were predicted to be present in oversaturation conditions, the calculation was repeated allowing precipitation.



### 3.3 Speciation of metal ions in fog

Chemical speciation of metal ions in fog was calculated at the measured pH value by means of the software PITMAP (Di Marco, 1998; Tapparo et al., 2020). Briefly, mass balance equations are solved, i.e. species concentration at equilibrium is obtained, using the Newton-Raphson method (Press et al., 2007). Speciation was done considering metal ions and ligands quantified in the fog samples: formate, acetate, propionate, oxalate, malonate, succinate, glutarate, maleate, methanesulphonate, $NO_3^-$, $Cl^-$, $SO_4^{2-}$, $PO_4^{3-}$, Mg(II), Ca(II), Sr(II), Ba(II), V(V, in the form $VO_2^+$), Cr(III), Mn(II), Fe(III),

Co(II), Ni(II), Cu(II), Zn(II), and Al(III). The input thermodynamic data (stoichiometry and stability constant of the complexes, including metal-aquo-complexes) were obtained from the literature (ScQuery v.5.84, 2005).

Unlike Visual MinteQ, PITMAP does not explicitly calculate the amount of precipitated solids. Speciation of metals obtained by Visual MinteQ and PITMAP for three fog samples (H603, H901, and H1002) was qualitatively equal with minor differences being observed only for species at very low concentrations ($<10^{-7}$ M) (see section S1 in SI).

## 4 Results and discussion

### 4.1 Meteorological conditions and fog occurrence

Figure 1 illustrates the meteorological conditions during the campaign, which were characterised by remarkable stability in terms of temperature (around 12 °C) with a narrow range (< 5°C) and humidity (RH 95%), which is a clear indication of the maritime influence. At the same time, a persistent stratocumulus cloud deck kept solar irradiance below 600 W m$^{-2}$ (Giorio

et al., 2022a).

Visibility data show the presence of fog events (classified as visibility <1 km) starting from 3 September 2017 and becoming persistent for extended periods from the night of 6-7 September 2017 (Figure 1). Two fog events were particularly intense over three consecutive days, the 9th, 10th, and 11th of September, with visibility reduced to below 100 m and liquid water content up to 0.3 g m$^{-3}$.





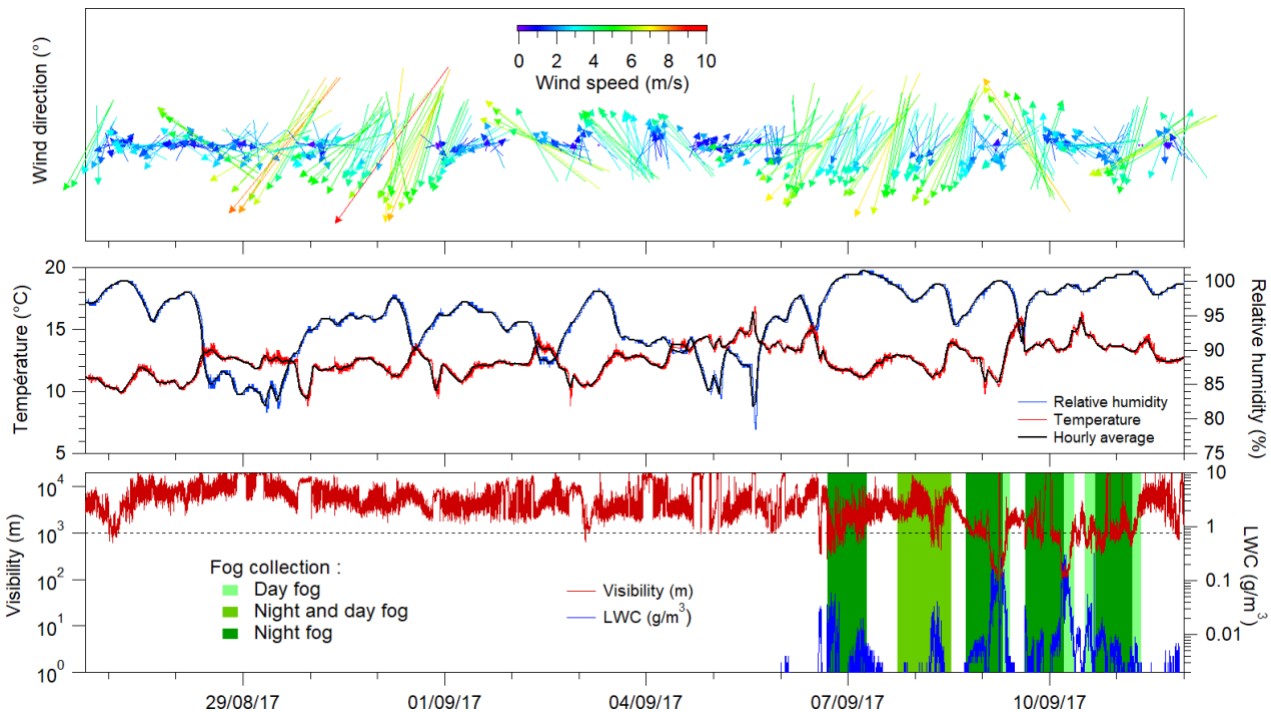

**Figure 1. Timeseries of wind direction and speed (top panel); B) RH%, T (middle panel); visibility and liquid water content (LWC) of the fog (bottom panel). The green shades correspond to the fog collection periods. Fog events are generally classified as visibility < 1 km.**

Wind patterns were characterised by a gentle (< 4 m s$^{-1}$) occasional land/sea breeze superimposed with moderate-to-strong (up to around 10 m s$^{-1}$) breezes coming predominantly from the NW and NNW directions and from the SW (Figure 1). Notable exceptions were the 4[th] of September (NE and NNW winds), the 5[th] of September (SWW winds), the 6[th] to 9[th] of September (SSW winds) and the 11[th] of September (SW winds). Air masses predominantly travelled along the coastline (NNW-SSE directions) before reaching the sampling site at Henties Bay. Those air masses often travelled above the ocean or the Namib desert, transporting sea spray, marine biogenic emissions, and mineral dusts. When winds were blowing from the S and SE directions, sporadic anthropogenic influences from the towns of Henties Bay and Walvis Bay could be expected (Giorio et al., 2022a). During fog events, the wind speed was almost always lower than 4 m s$^{-1}$ and from the SSW to the WNW direction. The observed fog was henceforth mostly due to the advection of marine stratocumulus or locally formed, confirmed by the fog microphysics (Weston et al., 2022) and chemical composition, as shown in the sections below.

**4.2 Fog chemical composition**

The major ions showed extremely high concentrations in fogs compared to other studies in coastal areas (Table 2), but in agreement with the findings of Gottlieb et al. (2019) for various sites in Namibia, while Fe, Zn and Cu concentrations were





of the same order or even reduced compared to other remote sites (Table 2) due to scarce vegetation sources in this Namib region.

Table 2 shows that the fogs were more diluted than local seawater in terms of major ions ($Na^+$, $Cl^-$, $K^+$, $NH_4^+$, $Mg^{2+}$, $Ca^{2+}$, $NO_3^-$, and $SO_4^{2-}$). This result is also corroborated by microphysics measurements that show rapid changes between fog, mist, and sea spray, the latter contributing significantly to reducing visibility (Formenti et al., 2019; Weston et al., 2022). Conversely, several compounds, such as Al, Ti, Ga, Fe, P and nitrate and all the rare earth elements (see Table S1 for elements not reported in Table 2) were more concentrated in coastal fog than in seawater. The elevated nitrate concentrations

can be attributed to the atmospheric processes affecting the important marine phytoplanktonic emissions of reactive nitrogen species (organic nitrogen and/or NO) ending in particulate nitrate formation (Altieri et al., 2021), henceforth explaining the presence of nitrate in the remote oceanic atmosphere. The correlation of the nitrate concentrations with oxalate and MSA to a lower extent ($R^2$=0.89 and 0.53, respectively) in our coastal fog samples supports this hypothesis. Carboxylic and dicarboxylic acids, notably highly oxidised species such as formate, acetate, succinate, and oxalate, also showed high

concentrations in fogs. A remarkable correlation between formate and acetate ($R^2 > 0.99$) suggests their similar sources or formation pathways. The high formate-to-acetate ratio (2.02) indicates the more important secondary formation for carboxylic acids in fog water, as compared to other possible sources (Li et al., 2020). Surprisingly, U was more concentrated in seawater than in fog. A source attribution analysis is summarised in the supplementary material in the form of enrichment factors (see Figures S4 and S5). Besides the local seawater, to represent the influence of the marine source, we also present

the enrichment factors calculated with respect to Al concentrations in the aerosol phase measured concurrently with the fog and representing the local dust content during the campaign (Formenti et al., this issue).

Generally, the chemical composition of the fog samples shows a large marine and mineral dust influence accompanied by highly processed organics which may have been formed *in situ* or in cloud then advected on the coast. Indeed, more and more studies show atmospheric VOC emissions from marine planktonic microorganisms. These

compounds can then undergo atmospheric photochemistry inducing the formation of new ultrafine particles which may grow in size and contribute to cloud condensation nuclei (Giorio et al., 2022a; Halsey et al., 2017; Mungall et al., 2017; Peltola et al., 2022).



**Table 2. pH, concentrations of major ions, organic acids, and elements measured in fogs and seawater at the coastal site at Henties Bay, and comparisons with previous coastal and marine studies. [a]** (Gottlieb et al., 2019) **(CM = coastal met, 15 km inland; VF = Vogelfederberg, 60 km inland); [b]** (Boris et al., 2018) and **Herckes (personal communication) for metals; [c]** (Benedict et al., 2012) **(VOCALS-Rex campaign); [d]** (Wang et al., 2014) **and** (Sorooshian et al., 2013) **for organic acids (E-PEACE campaign). "nss"=non-sea salt. Additional elements are reported in Table S1.**

| | Coastal fogs This study | | | | Coastal seawater This study | Namib fogs (2015-2016)[a] | | California fog and sea spray (2015) (Santa Barbara)[b] | | South-East Pacific (2008)[c] | Eastern Pacific (2011)[d] |
|---|---|---|---|---|---|---|---|---|---|---|---|
| | med | avg | min | max | avg | coastal fog (CM) | inland fog (VF) | coastal fog | sea spray | Marine clouds (avg) | Stratocumulus clouds (avg) |
| pH | 7.28 | 7.21 | 5.95 | 7.8 | 7.68 | | | 5.92 | | 4.4 | 4.46 |
| Na$^+$ (mM) | 81.6 | 76.6 | 9.40 | 184 | 432 | 41.49 | 8.70 | 0.101 | 35.21 | 0.53 | 0.39 |
| NH$_4^+$ (mM) | 1.58 | 1.86 | 0.19 | 5.73 | 9.51 | | | 0.232 | | 0.044 | |
| K$^+$ (mM) | 2.24 | 2.02 | 0.23 | 4.91 | 11.06 | 0.732 | 0.177 | 0.0102 | 0.797 | 0.012 | 0.00797 |
| Mg$^{2+}$ (mM) | 9.37 | 8.58 | 1.12 | 19.57 | 47.98 | 2.34 | 0.665 | 0.0141 | 3.90 | 0.060 | 0.049 |
| Ca$^{2+}$ (mM) | 2.57 | 2.62 | 0.34 | 7.56 | 11.32 | 3.57 | 1.38 | 0.0212 | 0.621 | 0.0285 | 0.012 |
| Cl$^-$ (mM) | 83.3 | 75.8 | 7.4 | 186 | 453 | 53.81 | 9.55 | 0.085 | | 0.460 | 0.57 |
| Nitrate (mM) | 0.24 | 0.33 | 0.04 | 1.15 | 0.087 | 1.08 | 0.564 | 0.126 | | 0.038 | 0.047 |
| Sulphate (mM) | 5.38 | 5.34 | 0.600 | 15.10 | 31.39 | 3.43 | 1.63 | 0.028 | | 0.070 | 0.050 (nss) |
| Phosphate (µM) | 15.5 | 24.9 | 13.4 | 63.3 | | | | | | | |
| Acetate (µM) | 33.24 | 47.80 | 7.25 | 219 | | | | 12.0 | | 0.208 | 3.9 |
| Propionate (µM) | 1.06 | 1.06 | 0.39 | 2.30 | | | | 1.40 | | | 0.64 |
| Formate (µM) | 59.35 | 86.47 | 12.52 | 430.3 | | | | 23.00 | | 0.392 | 3.1 |
| MSA (µM) | 4.44 | 5.20 | 0.86 | 19.44 | | | | 3.01 | | | 2.85 |
| Glutarate (µM) | 6.21 | 8.63 | 0.48 | 39.16 | | | | 1.81 | | | |
| Succinate (µM) | 13.05 | 23.05 | 0.56 | 146.8 | | | | 3.03 | | | |
| Malonate (µM) | 7.08 | 12.20 | 4.10 | 55.0 | | | | 4.16 | | | 4.1 |
| Maleate (µM) | 0.08 | 0.36 | 0.02 | 1.04 | | | | 0.52 | | | 1.2 |
| Oxalate (µM) | 16.2 | 39.0 | 5.88 | 214.0 | | | | 11.2 | | 0.1625 | 1.6 |
| Al (µM) | 36.51 | 35.96 | 9.84 | 64.25 | 2.89 | | | 2.51 | 19.2 | | 0.47 |
| P (µM) | 23.77 | 31.35 | 3.87 | 72.97 | 5.73 | | | | | | |
| Ti (µM) | 0.627 | 0.574 | 0.122 | 1.040 | 0.065 | | | | | | 0.0082 |
| V (µM) | 0.144 | 0.163 | 0.045 | 0.276 | 0.048 | | | 0.03 | 0.05 | | 0.06 |
| Cr (µM) | 0.790 | 0.886 | 0.159 | 2.321 | 0.011 | | | 0.0036 | 0.028 | | 0.0063 |
| Mn (µM) | 1.73 | 1.33 | 0.231 | 2.54 | 0.016 | | | 0.2 | 0.35 | 7.6 | 0.027 |
| Fe (µM) | 38.0 | 38.9 | 8.10 | 76.2 | 3.76 | | | 1.34 | 15.07 | 84 | 0.29 |




| | | | | | | | | | | | |
|---|---|---|---|---|---|---|---|---|---|---|---|
| Co (µM) | 0.070 | 0.071 | 0.014 | 0.117 | 0.001 | | | 0.00205 | 0.0078 | | 0.0033 |
| Ni (µM) | 2.48 | 3.42 | 0.771 | 6.85 | 0.018 | | | 0.01 | 0.0557 | | 0.019 |
| Cu (µM) | 0.388 | 0.699 | 0.055 | 1.95 | 0.046 | | | 0.1 | 0.11 | | 0.18 |
| Zn (µM) | 0.282 | 0.290 | 0.112 | 0.569 | 0.035 | | | 0.59 | 0.45 | | 0.11 |
| As (µM) | 0.035 | 0.037 | 0.009 | 0.063 | 0.022 | | | | | | 0.00059 |
| Sr (µM) | 9.78 | 11.33 | 1.77 | 20.64 | 72.0 | | | 0.06 | 4.63 | | 0.08 |
| Ba (µM) | 0.272 | 0.228 | 0.04 | 0.368 | 0.080 | | | 0.13 | 0.16 | | 0.0079 |
| Mo (nM) | 26.81 | 26.39 | 5.65 | 45.58 | 117.53 | | | 2.10 | 7.32 | | |
| Cd (nM) | 6.43 | 5.37 | 1.11 | 8.86 | 1.08 | | | 0.49 | 2.20 | | 310 |
| W (nM) | 0.78 | 50.42 | 0.31 | 299 | 0.29 | | | 0.20 | 0.30 | | |
| Pb (nM) | 14.1 | 11.6 | 1.70 | 18.6 | 0.89 | | | 1.80 | 14.80 | | 1.2 |
| U (nM) | 2.38 | 2.26 | 0.32 | 4.24 | 13.91 | | | | | | |

### 4.3 Comparison of metal solubility in coincident aerosol and fog samples

When soluble metal concentrations in TSP are compared with concentrations in fog (referred to as the total volume of the air parcel sampled) concurrently collected, higher concentrations of Al, Fe, and Ni are found in fog (Table 3). The Al, Fe, Cu, and Ni concentrations in fog are up to a factor of 4.7, 17, 5, and 6.5 higher than soluble concentrations in TSP, respectively. Similarly, Cr was up to 14.5 times higher than in the soluble fraction of TSP and Ti was detected in fog but not in the corresponding samples of the soluble fractions of TSP.

This result suggests that the chemical processing of airborne particles in fog would enhance metal solubility, such as the formation of metal-ligand complexes and/or photochemical processes if they are both locally produced and TSP act as condensation nuclei for fog formation. Previous studies have shown that Fe solubility can be driven by the formation of metal-ligand complexes with oxalate (Giorio et al., 2022b; Paris et al., 2011; Paris and Desboeufs, 2013). Similar processes can also impact Cu and Al solubility (Giorio et al., 2022b). Other organics, such as malonate, tartrate or HULIS (Paris and Desboeufs, 2013), can also influence Fe solubility. It is also the case of formate and acetate which, despite being weaker complexing agents, are better pH buffers (Upadhyay et al., 2011). The Fe solubility is strongly dependent on pH (Desboeufs et al., 1999, 2003), however, this pH dependency does not explain the observations in this study, as Fe is supposed to be more soluble at acidic pH rather than the neutral pH of the fog samples (Figure 2).

A previous study on the same site showed that Fe solubility in PM$_{10}$ was correlated with methanesulphonic acid concentrations, suggesting that photochemical reduction may be responsible for the enhanced Fe solubility (Desboeufs et al., 2024). Such a process would be compatible with the observations of a higher Fe solubility in fog samples, with the largest solubility differences observed in samples collected during daytime (Table 3). It is worth keeping in mind that LWC may be underestimated in nighttime fog due to the continuous running of the fog collectors until morning which leads to an overestimation of the volume of air sampled. Formenti et al., this issue show that fluoride could also play a role in Al, Ti, Si,





and Fe solubilities. However, as all fog samples were collected in a period of low fluoride concentrations, it can be excluded

that this species could play a role in the enhanced metal solubility in the fog samples.

**Table 3. Ratio between the mass concentration of dissolved metals in fog samples and soluble metals in the corresponding TSP**
**samples collected in Henties Bay during the AEROCLO-sA campaign in August-September 2017. Values of ratios for which metal concentrations in the soluble fraction of TSP were below detection limits while they were above detection limits in fog are reported as "nd" = not defined. Ratios above 1 indicate higher solubilities in fog and are highlighted in green.**

| Fog Sample ID | H603 | H801 | H901 | H902 | H1001 | H1002 | H1003 |
|---|---|---|---|---|---|---|---|
| TSP Sample ID | S-26 | S-31 | S-32 | S-33 | S-34 | S-34 | S-35 |
| Day/Night | Night | Night | Day | Night | Day | Day | Night |
| Al | 0.8 | 0.6 | 2.0 | 2.0 | 0.9 | 4.7 | 1.2 |
| Ca | 0.8 | 0.1 | 0.2 | 0.2 | 0.1 | 1.0 | 0.3 |
| K | 0.5 | 0.1 | 0.1 | 0.1 | 0.1 | 0.7 | 0.2 |
| Mg | 0.6 | 0.1 | 0.1 | 0.1 | 0.1 | 0.8 | 0.3 |
| Na | 0.6 | 0.1 | 0.2 | 0.2 | 0.1 | 1.1 | 0.3 |
| As | 0.2 | 0.1 | 0.2 | 0.2 | 0.1 | 0.8 | 0.2 |
| Ba | 0.6 | 0.1 | 0.3 | 0.2 | 0.1 | 0.9 | 0.2 |
| Co | 0.7 | 0.6 | 0.6 | 1.4 | 2.2 | 5.4 | 0.8 |
| Cr | 0.8 | 1.6 | 2.2 | 2.9 | 1.0 | 14.5 | 0.7 |
| Cu | 2.1 | 0.3 | 5.0 | 2.1 | 0.2 | 1.4 | 0.6 |
| Fe | 2.8 | 2.6 | 3.8 | 3.7 | 2.2 | 17.7 | 3.0 |
| Mn | 0.5 | 0.3 | 0.2 | 0.4 | 0.2 | 1.3 | 0.3 |
| Ni | 2.7 | 1.0 | 0.8 | 2.5 | 0.9 | 6.5 | 2.0 |
| P | 0.5 | 0.1 | 1.6 | 0.3 | 0.1 | 0.3 | 0.1 |
| Pb | nd | nd | nd | nd | nd | nd | nd |
| Sr | 0.6 | 0.1 | 0.2 | 0.2 | 0.1 | 0.8 | 0.3 |
| Ti | nd | nd | nd | nd | nd | nd | nd |
| V | <0.1 | <0.1 | <0.1 | <0.1 | <0.1 | 0.1 | <0.1 |
| Zn | 0.1 | <0.1 | 0.1 | <0.1 | <0.1 | <0.1 | <0.1 |

## 4.4 Speciation of metals in TSP vs. Fog

The speciation of the most abundant metals (see sections 0 and 0) in the deliquescent TSP as well as in the fog samples is

calculated to investigate which species are present in solution and could drive the solubility of each metal.

ALW content and pH, estimated using the model E-AIM IV, were in the range of 0.122-412 mg m$^{-3}$ and 0.1-3.2,

respectively (Figure 2). The high ALW indicates that the aerosol fraction, dominated by sea salt, was likely always in a



deliquescent state due to the mild temperatures and elevated RH (observed from September 6[th] until the end of the campaign

(Figure 1). Generally, the aerosol acidity was inversely proportional to the ALW content. It is worth noticing that the

calculated aerosol pH is very low (pH < 3) compared to the fog samples (neutral pH) concurrently collected. The measured

pH values for the fog samples ranged from 5.95 to 7.8, with a median value of 7.28, similar to that of seawater (Table 2).

These pH values are toward the upper end of values reported in the literature, which, however, refer to urban and agricultural

areas (Ge et al., 2022; Gundel et al., 1994; Paglione et al., 2021; Pye et al., 2020; Wen et al., 2005). The higher pH measured

in this study in fog samples is likely due to a high water content as well as the dominance of the sea spray component. The

pH in fog might be at least partly driven by the presence of carbonates, as evidenced by the moderate correlation between -

$\log([Ca(II)])$ and pH ($r^2 = 0.21$), similar to what happens in seawater (Jiang et al., 2019). The E-AIM IV model does not

consider K(I), Mg(II), Ca(II), and carbonates, and, therefore, may underestimate aerosol pH compared to what is expected

for sea spray aerosol (Pye et al., 2020).

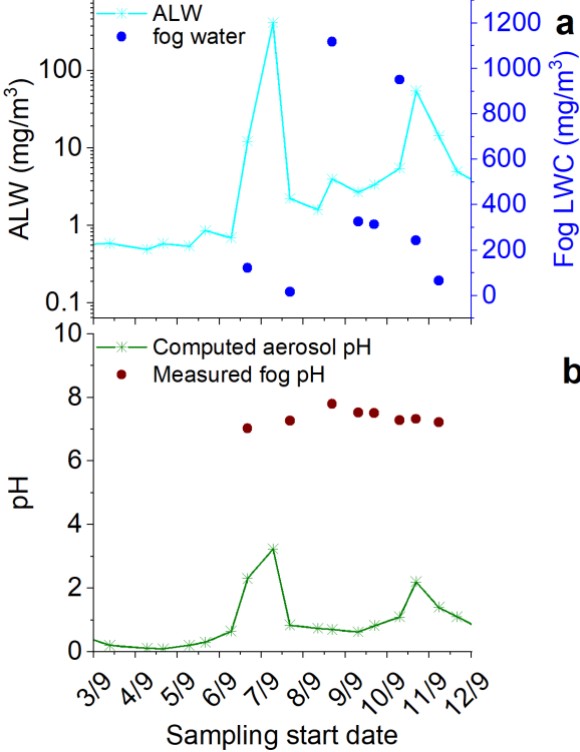


**Figure 2. Timeseries of (a) aerosol liquid water content (ALW) calculated with E-AIM and measured fog liquid water content (LWC) retrieved from the measured water collected with the CASCC sampler and, (b) pH in aerosol calculated with E-AIM and measured pH in fog. Fog samples H1001 and H1002 have been averaged over the TSP sample S-34 collection time.**

Speciation of metal ions in deliquescent aerosol depended on ALW content, pH, and the amount of inorganic and organic

ligands in the samples. Due to the high ALW and low pH, a large fraction of the metal ions is in free form, with such a



fraction being dominant during the second part of the campaign characterised by higher RH and fog events. For most of the metals, another significant fraction is bound to inorganic compounds such as $SO_4^{2-}$ for Al(III), Cr(III), Cu(II), and Ni(II), and $Cl^-$ for Ca(II), Mg(II), Mn(II), and Zn(II). Fe is the only exception with Fe(II) being equally distributed between binding

$SO_4^{2-}$ and $Cl^-$, and Fe(III) being predominantly complexed with oxalate (Figure S6). Besides Fe(III), a significant fraction of Al(III) and a minor fraction of Cu(II) are also predicted to bind oxalate. In addition, in the first part of the campaign when ALW was relatively lower, part of Ca(II) is calculated to precipitate as $CaSO_4$. Given that pH was likely underestimated in our calculations, the real speciation picture may be shifted in favour of a larger proportion of hydroxide formation for some metal ions.

Fog samples were characterised by much higher water content, ranging from 16 to 1,117 mg m$^{-3}$, and close to a neutral pH. Given the extremely different pH conditions, we would expect the speciation picture to be completely different compared to that of TSP.

The most striking and surprising difference is observed for Al(III) and Fe(III), which, according to our calculation, would form insoluble hydroxides in fog water (Figure 3) and therefore should not be in solution, unless they are present in a

colloidal form (Shi et al., 2009). This result runs counter observations from ICP-MS analysis which shows an even higher concentration of Al and Fe in solution in the fog samples compared to the soluble concentration in TSP. In the case of Al(III), only a minor component is predicted to be bound with organics such as oxalate and succinate (Figure 4b). Ni(II) speciation in fog shows similar results compared with TSP (Figure 3) with the only difference being that Ni(II) is partly present as soluble hydroxide in fog in addition to being present in free form and bound to $SO_4^{2-}$ as in TSP. Only a minor

component of Ni(II) is predicted to be bound to organics such as oxalate, malonate, and succinate (Figure 4b). A notable difference between the two types of samples is observed instead for Cu(II) which becomes bound to oxalate and malonate for a fraction between 19 and 43% (Figure 4a) in the fog samples while in the TSP only a minor fraction (<7.3%) was bound to organics. In the case of Cr(III), speciation in TSP showed an almost equal distribution between free form and bound to $SO_4^{2-}$ for the only two samples in which Cr was detected in the soluble fraction, while in fog it is mainly present as soluble

hydroxide and mixed $PO_4^{3-}$-hydroxide complex (Figure S7).



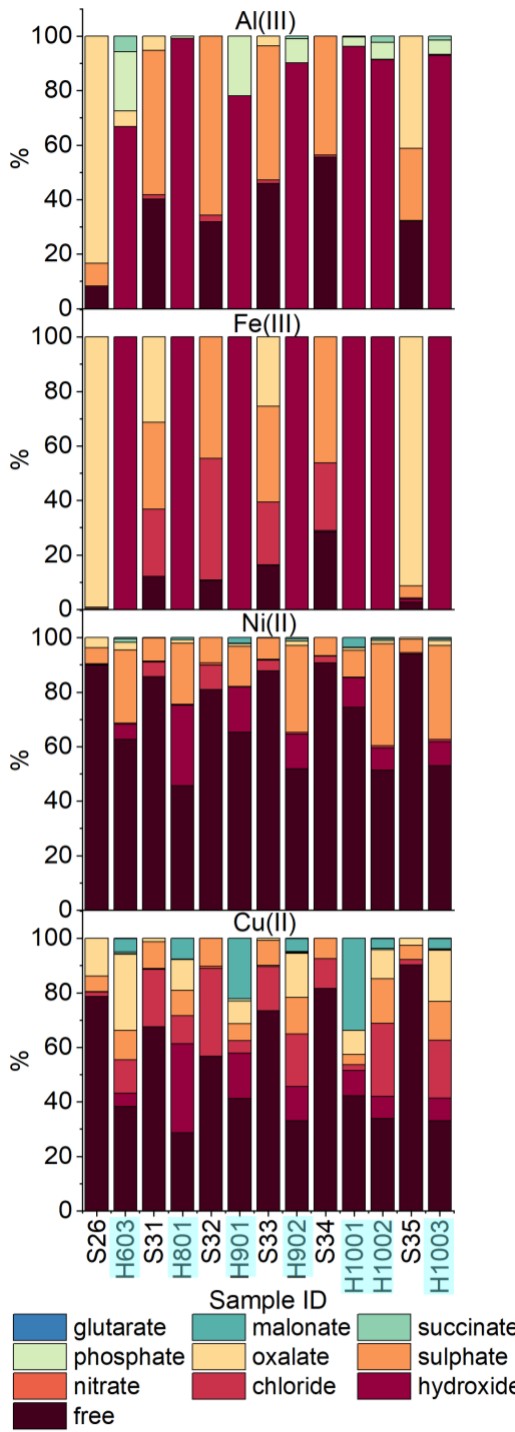

**Figure 3. Comparison of the detailed speciation of soluble metals in TSP (sample IDs starting with "S") and in fog (sample IDs starting with "H") for (top-to-bottom) Al(III), Fe(III), Ni(II), and Cu(II). Samples collected at approximately the same time are shown side-by-side. The sampling of TSP sample S34 covered the same times as both "H1001" and "H1002" fog samples.**





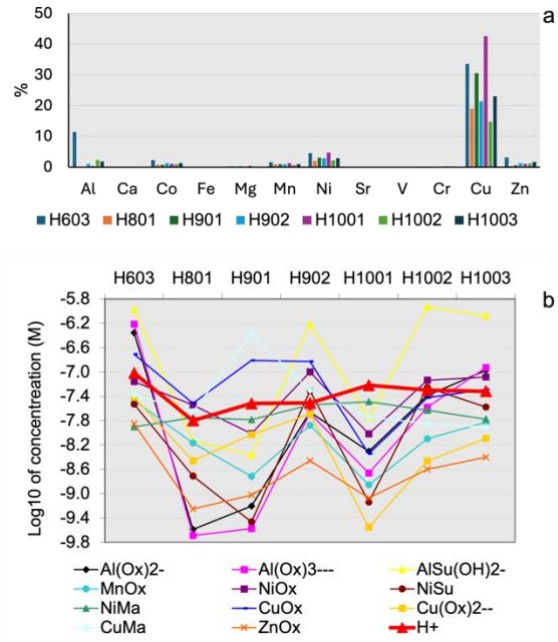

**Figure 4. Contribution (in %) or organic complexation to the total concentration of metals in the fog samples (a) and concentrations of the main metal-organic ligand complexes (b). Ox=oxalate, Su=succinate, Ma=malonate.**

**4.5 Investigation of contrasting Fe solubility in TSP and fog using XANES and TEM**

Iron speciation was measured experimentally in order to elucidate the processes that may increase iron solubility in the fog samples compared to the TSP samples. Table 4 and Figure 5 show the results of the linear combination fitting of the XANES spectra of the TSP and fog samples using reference spectra that are representative of various iron oxides, clays, pyrite, and complexes with oxalate formed either by Fe(II) or Fe(III). The results show that for both TSP and fog samples, iron oxides and clays are by far the major components, as expected for samples affected by the transport of mineral dust. Iron-oxalate complexes, in particular those of Fe(III), contributed to some samples of both TSP and fog. Excluding samples that were close to the detection limit or affected by diffraction, iron-oxalate contributed up to 6% in TSP samples and up to 8% in fog samples. Pyrite was found to contribute up to 7.8% of the total iron concentration, which was attributed to the transport of fugitive dust from smelting areas (Formenti et al., this issue).

The presence of a significant amount of iron-oxalates in more than one fog sample (Table 4) could explain the enhanced solubility of iron in fog compared to TSP samples. This amount is comparable with the soluble fraction of iron (< 2% in TSP in these samples). The results of the XANES measurements run counter the results of the thermodynamic modelling that predicted iron to precipitate in fog samples in the form of hydroxides but they are consistent with ICP-MS measurements that showed that indeed soluble Fe was present in the fog samples. On the other hand, the thermodynamic modelling results



are qualitatively consistent with XANES measurements on TSP samples, for which oxalate of iron complexes were expected and indeed experimentally detected. We hypothesise that iron-oxalates complexes are formed through aqueous phase processing of iron during particle activation into droplets when the water content is lower and pH is also lower than in the collected fog samples. These complexes may then remain in the fog droplets in a kinetically stable form before being converted to insoluble hydroxides. Additionally, as activation to droplet is a size-dependent process, there may be a fractionation of soluble iron whereby, for example, smaller particles with a higher soluble iron content may contribute more to fog droplets compared to larger particles with a lower soluble iron content (Marcotte et al., 2020). The XANES measurements also showed that iron was predominantly present in the (III) oxidation state. This result would suggest that photochemical reduction to Fe(II) was not a significant process involved in the enhanced iron solubility in the fog samples.

Besides organic iron formation during droplet activation, colloidal iron may be present in fog (Gledhiir and Buck, 2012) that would be detected as soluble in ICP-MS measurements. To test this hypothesis, TEM and DLS measurements were performed on fog samples before and after filtration through a 0.22 μm pore-size filter.

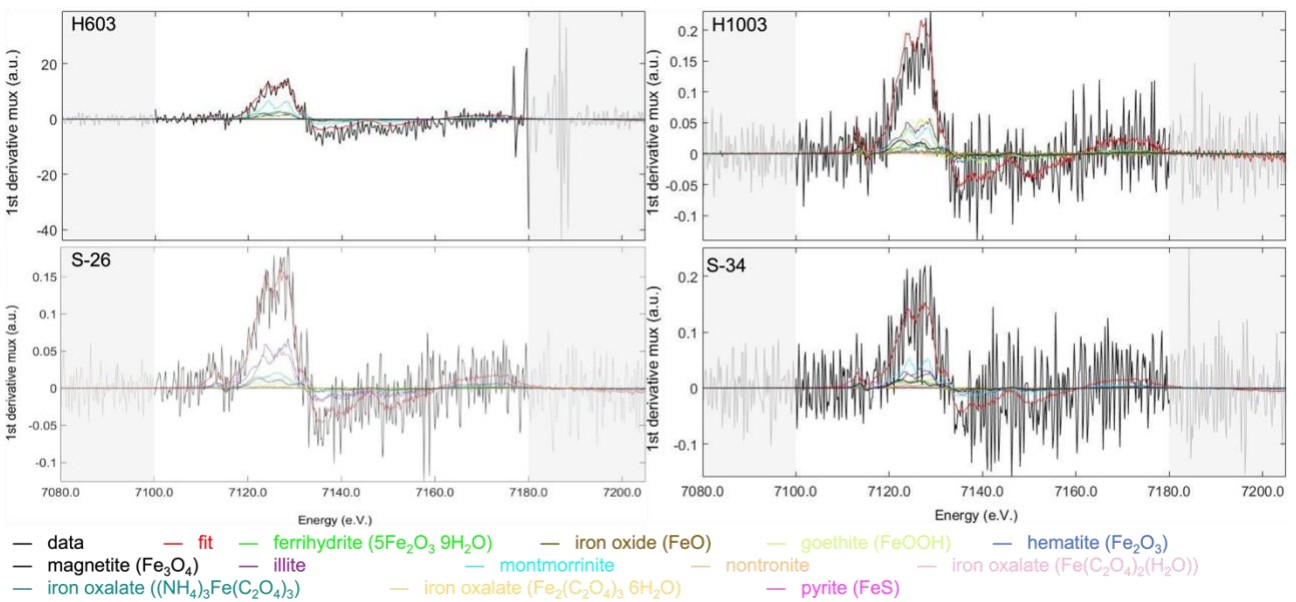

**Figure 5. XANES spectra at the Fe K-edge for two fog samples (top, H603 and H1003) and two TSP samples (bottom, S-26 and S-34) showing the results of the spectral deconvolution using a linear combination fitting with Fe measured reference standards of oxides, clays, organic complexes and sulphide.**



**Table 4. Results of the spectral (XANES, Fe K edge) deconvolution done on TSP and Fog samples using a linear combination fitting with Fe reference standards of iron oxides, clays, organic complexes and sulphide. Results are reported as % contribution of each component used for the linear combination fitting.**

| | TSP S26 | Fog H603 | TSP S28 | Fog H701 | TSP S31 | Fog H801 | TSP S32 | Fog H901* | TSP S33 | Fog H902 | TSP S34 | Fog H1001* | TSP S35* | Fog H1003 | TSP S36* | Fog H1101 |
|---|---|---|---|---|---|---|---|---|---|---|---|---|---|---|---|---|
| Pyrite | 5.7 | 0 | 3.3 | 0 | 5.0 | 2.5 | 7.7 | 0 | 5.6 | 4.7 | 7.3 | 0.1 | | 3.0 | | 7.8 |
| Illite | 34 | 11.3 | 16.8 | 33.1 | 29.8 | 40.4 | 25.4 | 8.3 | 16.9 | 33.1 | 15.3 | 10.2 | | 22.4 | | 20.6 |
| Montmorrinite | 8.1 | 43.5 | 15.5 | 31 | 21.6 | 17.4 | 26.6 | 17.3 | 20.2 | 28.1 | 28.2 | 10.4 | | 17.4 | | 12.5 |
| Nontronite | 0 | 0 | 5.7 | 0 | 0 | 3.3 | 0 | 16.9 | 7.7 | 0.4 | 3.7 | 11.1 | | 9.6 | | 0.9 |
| Total iron clays | 42.1 | 55 | 38 | 64 | 51.4 | 61 | 52 | 43 | 44.8 | 62 | 47.2 | 32 | | 49 | | 34 |
| ferrihydrite | 11.5 | 18.9 | 16.5 | 8.8 | 21.3 | 7.9 | 15.3 | 15.7 | 19 | 1.4 | 8.4 | 10.6 | | 5.4 | | 17.8 |
| FeO | 1.0 | 0.1 | 0 | 0 | 2.3 | 0 | 2.6 | 0 | 0 | 1.5 | 0 | 6.0 | | 2.0 | | 6.0 |
| Goethite | 3.2 | 8.6 | 4.9 | 15.2 | 0 | 5.4 | 4 | 12.1 | 6.2 | 15.1 | 7.6 | 12.1 | | 20.5 | | 7.2 |
| Hematite | 26.6 | 17.2 | 12.5 | 4.4 | 3.9 | 10.9 | 3.3 | 3.6 | 5.3 | 5.4 | 18.8 | 8.4 | | 1.3 | | 4.5 |
| Magnetite | 8.8 | 0 | 18.5 | 7.5 | 14.4 | 6.3 | 13.8 | 26 | 19 | 2.2 | 10.3 | 6.8 | | 10.9 | | 4.4 |
| Total iron oxides | 51.1 | 45 | 52.4 | 36 | 41.9 | 31 | 39 | 57 | 49.5 | 26 | 45.1 | 44 | | 40 | | 40 |
| $Fe(C_2O_4)2H_2O$ | 0 | 0.3 | 0 | 0 | 0 | 0 | 0 | 0 | 0.1 | 1.9 | | 22.3 | | 1.4 | | |
| $(NH_4)_3[Fe(C_2O_4)_3]$ | 0 | 0 | 0 | 0 | 1.7 | 2.2 | 0 | 0 | 0 | 6.1 | 0.5 | 1.7 | | 5.0 | | 11.9 |
| $Fe_2(C_2O_4)_3 6H_2O$ | 1.3 | 0 | 6.2 | 0 | 0 | 3.7 | 1.3 | 0 | 0 | 0 | | 0.3 | | 1.0 | | 6.3 |
| Total organic iron | 1.3 | 0 | 6.2 | 0 | 1.7 | 6 | 1.3 | 0 | 0.1 | 8 | 0.5 | 24 | | 7 | | 18 |

*noisy spectra, affected by diffraction





**Figure 6. Results of the analyses of filtered and sedimented fog samples showing (a) the normalised distribution of hydrodynamic diameters of suspended particulates, weighted by volume, from DLS measurements, (b) an elemental map of a nanoscopic suspended particles, with the scale bar representing 50 nm, imaged by TEM and (c) an energy dispersive X-ray spectra showing the relative intensity of elements occurring in the particle mapped in "b".**

Particles identified by DLS in the sub-100 nm range, after filtration and sedimentation, reveal the presence of a colloidal phase within the fog samples (Figure 6a). TEM confirmed this hypothesis, as distinct particles with diameters ranging from 50-150 nm were identified (Figure 6b). These are likely to be the same particles observed using DLS. EDS showed that these particles had cores containing silicon, aluminium, iron and sulphur, among other less prominent elements (Figure 6c).




## 5 Conclusions

In this study, we compared TSP and fog samples collected in parallel on the Namibian coast at Henties Bay during the AEROCLO-sA campaign in September 2017.

We found contrasting element solubility in TSP and fog samples, with higher than expected solubility of Al, Fe, Cu, Ni, and Cr in fog. Thermodynamic modelling of aqueous phase equilibria in deliquescent aerosols and fog suggests that Cu solubility in fog could be due to complexation with inorganic (i.e., chloride and sulphate) and organic ligands (i.e., oxalate and

malonate) which is favoured by the neutral pH observed in fog samples. For Cr, the enhanced solubility in fog could be due to the formation of soluble hydroxides and phosphate-hydroxide complexes in fog. For Ni, the thermodynamic modelling shows that it may be present partly in free form, partly as soluble hydroxide, partly as sulphate, and only a minor component would be complexed by organics in fog.

Previous studies have shown that aqueous phase processing can lead to an increase in the solubility of various elements, such

as iron and aluminium (Giorio et al., 2022b; Li et al., 2024; Shi et al., 2015; Tapparo et al., 2020). However, the majority of observations report an acidic pH for fogs/clouds (Pye et al., 2020) which would be consistent with increased element solubility. Here we show enhanced solubility of Al and Fe in fog characterised by a neutral pH, in which these elements would be predicted to precipitate as hydroxides. The thermodynamic modelling results run counter experimental measurements that show that both Al and Fe are detected in fog (by ICP-MS) and that Fe can be present as an oxalate

complex in fog (from XANES measurements) as well as in colloidal form (from DLS and TEM measurements).

We hypothesise that: 1) Al and Fe undergo aqueous phase processing during particle activation into droplets, when the water content and pH are lower, and the formation of metal-ligand complexes with organic acids are favoured, 2) Al and Fe remain in fog water as kinetically stable metal-organic complexes or as a colloidal suspension of nanoparticles.

Our study considers the bulk composition of TSP and fog samples for thermodynamic calculations, however, single particle

composition may be very different from the bulk composition. The single particle composition would be the most relevant for understanding the processing happening during particle activation into droplets as the activation can be both size- and composition-dependent (Marcotte et al., 2020; Zhang et al., 2024). Nevertheless, the finding that Al and Fe solubility can be enhanced in fog even at neutral pH expands the breadth of the importance of aqueous phase processing in the atmosphere. The enhanced solubility of Al and Fe, as well as their complexation with organics, can have impacts on their ability to

produce reactive oxygen species (Campbell et al., 2023; Shahpoury et al., 2021, 2024a, b) as well as potentially modifying particle optical properties.

## Data Availability

All data are made freely available by the French national service for atmospheric data AERIS-SEDOO data at https://baobab.sedoo.fr/AEROCLO/. The FASTOSH XANES data analysis package is available for download at

https://www.synchrotron-soleil.fr/fr/lignes-de-lumiere/samba (last accessed: 02/03/2024).



## Author contribution

CG and PF conceived the study, analysed data, and drafted the manuscript. CG, KD, and PF collected TSP samples. AM collected and analysed fog samples. PH and DN analysed fog samples. GL, PF, FB, and CDB performed XANES measurements. DW performed TEM and DLS measurements. CG, VDM, MN and SD performed thermodynamic modelling
calculations. ST, SC, and KD analysed TSP samples. FB provided visibility data. JFD, SJP, and AN contributed to sample and data collection and provided support for the field campaign. All authors revised and approved the manuscript before submission.

## Special issue statement

This article is part of the special issue "New observations and related modelling studies of the aerosol–cloud–climate system
in the Southeast Atlantic and southern Africa regions (ACP/AMT inter-journal SI)". It is not associated with a conference.

## Competing interests

The authors declare that they have no conflict of interest.

## Acknowledgements

Authors are grateful to the AEROCLO-sA consortium for their work in the field and during the preparation of the field
campaign, and to SANUMARC for hosting the field campaign. Authors are grateful to Andrea Osti for help with the calculations using PITMAP. The authors wish to thank AERIS (https://www.aeris-data.fr/), the French centre for atmospheric data and service, for providing the campaign website and organising the curation and open distribution of AEROCLO-sA data. Finally, SOLEIL is acknowledged for provision of synchrotron radiation facilities (Proposal 20210043).

## Funding support

This work was supported by the French National Research Agency under grant agreement n° ANR-15-CE01-0014-01, the French national programme LEFE/INSU, the French National Agency for Space Studies (CNES), and the South African National Research Foundation (NRF) under grant UID 105958. CG's work was supported by the Supporting TAlent in ReSearch@University of Padova STARS-StG "MOCAA", and a BP Next Generation fellowship awarded by the Yusuf
Hamied Department of Chemistry at the University of Cambridge.



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
