# Peer review of "Contrasting solubility and speciation of metal ions in total suspended particulate matter and fog from the coast of Namibia"

_EGUsphere, 2024_

## Author Comment (AC1)

The authors are grateful to anonymous reviewer 1 for providing feedback to the manuscript. Our detailed point-by-point response to the comments is reported below.

Reviewer 1
This study investigates the differences in metal concentrations between TSP (total suspended particulate) and fog samples collected concurrently at a coastal site in 2017. Using ICP-MS analysis and thermodynamic modeling, the authors attribute these variations to distinct complexation behaviors with inorganic/organic ligands. Key findings include enhanced Al/Fe solubility in fog water under neutral pH conditions, hypothesized to result from aqueous-phase processing during droplet activation. XANES analysis confirmed Fe-organic complexes despite thermodynamic predictions favoring hydroxide species. TEM/DLS measurements further supported colloidal Fe nanoparticles as an additional phase. I have the following comments.

Major comments:

1. The manuscript does not explicitly state whether total or water-soluble metals were measured via ICP-MS. Metal solubility should be quantified as the soluble fraction relative to total metal content. Equating bulk concentration to "solubility" is problematic. A revised discussion addressing this distinction is needed.

With ICP-MS we measured the dissolved fraction of the metals, that is from samples extracted in ultrapure water and filtered with a 0.2 um filter. The methodology for the chemical analysis is reported in the companion paper by Formenti et al. which is now published in EGUsphere (https://doi.org/10.5194/egusphere-2025-446). The discussion reported in this manuscript always refers to the dissolved fraction while information on the solubility of the metals (in % over the total) is reported in the companion paper.

2. The method section lacks critical information for ICP-MS analysis. The manuscript only has one sentence for this. E.g. how are the samples prepared? What are the instrumental parameters. This affects how the metal data should be interpreted.

For TSP samples, all information can be found in the companion paper (Formenti et al., this issue).

For fog samples, the following text has now been added at line 147: "For dissolved trace metal determination, fog samples were aliquoted in the field into precleaned (with nitric acid) HDPE bottles and stored at 4°C until analysis. Prior to analysis, samples were filtered through a 0.22 µm filter (PES, Millipore) and acidified using high purity nitric acid (EMD Chemicals, 67–70% Omni Trace Ultra) to a pH less than 2. The samples were analysed for 40 trace metals using a Thermo Electron X-Series 2 quadrupole inductively coupled plasma mass spectrometer (ICP-MS). All elements were analysed using Kinetic Energy Discrimination (KED) mode using as collision cell gas a hydrogen (7%) and helium mixture. Scandium, germanium, yttrium, indium and bismuth were used as internal standards. As for TSP samples, the dissolved fraction is defined operationally as passing through a 0.2 µm filter which may include nano-sized particles as reported in a previous study (Giorio et al., 2025)."

Reference: Giorio, C., Borca, C. N., Zherebker, A., D'Aronco, S., Saidikova, M., Sheikh, H. A., Harrison, R. J., Badocco, D., Soldà, L., Pastore, P., Ammann, M., and Huthwelker, T.: Iron Speciation in Urban Atmospheric Aerosols: Comparison between Thermodynamic Modeling and Direct Measurements. ACS Earth and Space Chemistry, 9 (3), 649–661, https://doi.org/10.1021/acsearthspacechem.4c00359, 2025.

3.  Current spectral plots in Figure 5 are challenging to interpret due to overlapping lines/symbols and low contrast colors/lineshapes. Consider stacked panels or splitting into subfigures for key comparisons. For example, it is hard to tell from the figure that Fe was predominantly present in the (III) oxidation state. Additionally, descriptions on how the simulations were done (i.e. parameters) should be added to the method section.

We have now improved Figure 5 following the suggestion for the reviewer.

[Figure]

**Figure 1. XANES spectra at the Fe K-edge for two fog samples (top, H0603 and H1003) and two TSP samples (bottom, TSP 26 and TSP 34) showing the results of the spectral deconvolution using a linear combination fitting with Fe measured reference standards of oxides, clays, organic complexes and sulphide. The individual contributions of the references to the total fit (red curve) are represented as stacked black lines. For sake of clarity, the contribution of organic complexes (named "oxalate 1" for $Fe(C_2O_4)2H_2O$, "oxalate 2" for $(NH_4)_3[Fe(C_2O_4)_3]$ and "oxalate 3" for $Fe_2(C_2O_4)_36H_2O$ in the Figure) are multiplied by 10.**

The description of the fitting procedure is reported in the companion paper from Formenti et al., this issue. "for Fe(II)-bearing minerals, the position of the centroid of the pre-edge is found at 7112.1 eV, whereas it is at 7113.5 eV for Fe(III)-bearing minerals." and "The speciation of Fe was obtained by the least-square fit of the measured XANES spectra based on the linear combination of mineralogical references. Fits were conducted on the first derivative of the normalized spectral absorbance in the energy region between 7100 to 7180 eV, corresponding to -30 and +50 eV of the K-edge. Only the fits with a $\chi^2$ closest to 1 were retained for further

analysis." Information on the standard reference compounds is reported in the supplements of Formenti et al., in Table S2. The companion paper is now available as pre-print in EGUsphere (https://egusphere.copernicus.org/preprints/2025/egusphere-2025-446/).

4. The discussion on pH and metal solubility is largely lacking. It is known that acidification of metal species in the presence of sulfate is an important mechanism to make insoluble metal become soluble. The author should consider adding relevant discussions. It would be useful to plot the pH against/with metal solubility.

Metal solubility in TSP is discussed in the companion paper by Formenti, et al. We have now explained this at the beginning of section 4.3: "Solubility of metals in TSP is discussed for the whole campaign in the companion paper by Formenti et al., 2025, this issue. Here we focus the discussion on the comparison between dissolved metals in pairs of TSP and fog samples."

Concerning the extraction protocols, while fog pH was measured and was always close to neutral, for TSP dissolved metal determination autogenic pH was used which may be more acidic (not measured). Despite this difference, that would cause a higher dissolved fraction for TSP compared to fog samples, we actually observed the opposite for most metals/samples and therefore it should not affect our conclusions. This difference has now been clarified in the methodology section at line 154: "One difference between the dissolved fraction in fog and in TSP may arise from the pH of the extraction solution, which was in both cases autogenic but may be more slightly more acidic for TSP (not measured) compared to fog samples (measured to be close to neutral)."

Minor comments:

1. Provide the year for this reference: "Formenti et al, this issue". This appears many times in the manuscript.

The reference has been updated and added to the reference list.

2. "…is discussed in section 0." Where is section 0? This also appears many times in the manuscript.

It looks like this is due to an issue in converting the word file to PDF. We have fixed this in the revised manuscript.

3. Revise single-sentence paragraphs at e.g. Lines 91/125/141 by integrating them logically into adjacent sections or expanding context where appropriate.

The sentences have been revised as follows:

Line 91: "The list of aerosol samples and concomitant fog samples relevant to this paper is presented in **Error! Reference source not found.**, while in the following paragraphs, we describe the strategy of fog and seawater collection and analysis."
Line 125, describing the liquid water content measurements with the CASCC has been moved to the previous section, where CASCC collection is described.
Line 141 (147 in the revised version) has now been modified as reported in the answer to comment 2.

4. In section 4.1, "temperature with a narrow range (<5°C) and humidity (RH 95%)". Provide a range for humidity.

We have now added the requested information: "RH 95 $\pm$ 4 %, min 78.5 %, max 100 %".

5. Section 4.3, line 282, "higher concentrations of Al, Fen and Ni.." Missing Cu here?

We have now added Cu.

6. Define LWC upon first use.

Definition added.

---

## Author Comment (AC2)

The authors are grateful to reviewer 2, Akinori Ito, for providing feedback to the manuscript. Our detailed point-by-point response to the comments is reported below.

Reviewer 2:
General comments

Model predictions of metal speciation and solubility in aerosols and fog are highly uncertain. The authors collected total suspended particulate (TSP) and fog water samples at Henties Bay and seawater on the seashore during the period of 3-11 September. They combined thermodynamical modelling with field measurements using X-ray absorption near edge spectroscopy (XANES), Transmission electron microscopy (TEM), Dynamic Light Scattering (DLS), and Inductively Coupled Plasma Mass Spectrometry (ICP-MS). Previous study at the Henties Bay Aerosol Observatory (HBAO) showed strong link between MSA (methane sulfonic acid) and iron solubility up to 20% but not between oxalate and iron solubility. Conversely, their XANES results showed that photochemical reduction to Fe(II) was not a significant process involved in the enhanced iron solubility in the fog samples but the presence of iron-oxalate complexes that could explain its enhanced solubility in fog samples. Their TEM and DLS measurements revealed the presence of nano-sized colloidal particles containing Fe and Al in filtered fog samples that may appear as soluble in ICP-MS measurements. The modeling exercises with direct measurements performed in this paper may help us to advance our understanding of metal speciation and solubility in aerosols and fog. I have some comments and questions to improve this paper.

Specific comments

l.61: Please specify the photo-reduction processes of marine biogenic emissions and explain the role of MSA (methane sulfonic acid) and oxalate in enhanced solubility to elucidate the complement to those findings.

We have now expanded this based on the reviewer's suggestion: "Such processes can involve the formation of photoactive complexes involving Fe(III) and methanesulphinic acid which absorbs and photolyses in the visible range to produce Fe(II) and methanesulphonic acid among other species (Siefert et al., 1994). Similarly, Fe(III)-carboxylate complexes can undergo photochemical redox reactions where the Fe(III) is reduced to Fe(II) (Johansen and Key, 2006).

Johansen, A. M., & Key, J. M. (2006). Photoreductive dissolution of ferrihydrite by methanesulfinic acid: Evidence of a direct link between dimethylsulfide and iron-bioavailability. *Geophysical Research Letters*, *33*(14). https://doi.org/10.1029/2006GL026010
Siefert, R. L., Erel, Y., & Hoffmann, M. R. (1994). Iron photochemistry of aqueous suspensions of ambient aerosol with added organic acids*. In *Pergamon Geochimica et Cosmochimica Acta* (Vol. 58, Issue 15).

l.68: What is the role of butenes in the formation of metal-ligand complexes?

There isn't a clear link between butenes and complex formation. We realised that the sentence can be misleading so we have removed the reference to butene emissions.

l.83, l.310: Please correct section 0.

We have corrected all the references to sections in the revised manuscript.

l.89: What are the differences in the leaching method from the previous study at HBAO? Please explain the leaching protocol whether nano-sized colloidal particles are measured as soluble in ICP-MS measurements.

The leaching method used here was the same as in previous studies, and described in the companion paper by Formenti et al., this issue. All samples analysed by ICP-MS were previously filtered through a 0.2 um filter that is not able to capture all nano-sized particles as shown in our recent paper (Giorio et al., 2025, Iron Speciation in Urban Atmospheric Aerosols: Comparison between Thermodynamic Modeling and Direct Measurements. *ACS Earth and Space Chemistry*. https://doi.org/10.1021/acsearthspacechem.4c00359). It is therefore the dissolved fraction of metals that was analysed by ICP-MS, comprising soluble metals and nano-sized colloidal suspensions. The word "soluble" has now been replaced with "dissolved" throughout the manuscript. We have also clarified this in the methodology section at line 153 of the revised manuscript: "As for TSP samples, the dissolved fraction is defined operationally as passing through a 0.2 μm filter which may include nano-sized particles as reported in a previous study (Giorio et al., 2025)."

l.108: Please explain the differences in the operational definition of "soluble" and "dissolved" to elucidate dissolved metals in fog and soluble metals in TSP. Please specify whether dissolved or soluble trace metals in your method include nano-sized colloidal particles.

This has now been addressed as a result of the previous comment.

l.161: Why don't you include organic complexation of Fe with humic-like substances, as you mentioned in l.290?

We didn't include complexes with humic-like substances due to a lack of reference standards.

l.193: This is probably the major reason of low pH calculated in sea salt and Ca-containing aerosols under high RH and thus resulting in free form for a large fraction of the metal ions as, as you mentioned in l.322. It is highly recommended to use the model which considers the alkaline minerals.

Unfortunately, E-AIM does not consider Ca and Mg. Isorropia II could be used, but various studies have pointed out that isorropia does not perform well when run in reverse mode (with only particle phase concentrations available) and at very low or very high relative humidities (such as in this campaign) (Hull et al., 2025; Haskins et al., 2018; Peng et al., 2019). In addition, we do not have measurements of carbonates that would also contribute to the uncertainty on the calculated pH. This has now been explained in the text at lines 201 of the revised manuscript: "This can lead to an underestimation of the real pH of aerosols. The model ISORROPIA II could be used to consider these species, but a few studies have pointed out that ISORROPIA does not perform well at very high relative humidity (such as in this campaign) when run in reverse mode (i.e., when only particle phase measurements are available) (Haskins et al., 2018; Hull et al., 2025; Peng et al., 2019). In addition, the absence of carbonate measurements can contribute to the uncertainty around the real aerosol pH."

Haskins, J. D., Jaeglé, L., Shah, V., Lee, B. H., Lopez-Hilfiker, F. D., Campuzano-Jost, P., Schroder, J. C., Day, D. A., Guo, H., Sullivan, A. P., Weber, R., Dibb, J., Campos, T., Jimenez, J. L., Brown, S. S., & Thornton, J. A. (2018). Wintertime Gas-Particle Partitioning and Speciation of Inorganic Chlorine in the Lower Troposphere Over the Northeast United States and Coastal

Ocean. *Journal of Geophysical Research: Atmospheres*, *123*(22), 12,897-12,916. https://doi.org/10.1029/2018JD028786

Hull, T., D'Aronco, S., Crumeyrolle, S., Hanoune, B., Giammanco, S., la Spina, A., Salerno, G., Soldà, L., Badocco, D., Pastore, P., Sellitto, P., & Giorio, C. (2025). Metal speciation of volcanic aerosols from Mt. Etna at varying aerosol water content and pH obtained by different thermodynamic models. *Environmental Science: Atmospheres*, *5*(1), 8–24. https://doi.org/10.1039/D4EA00108G

Peng, X., Vasilakos, P., Nenes, A., Shi, G., Qian, Y., Shi, X., Xiao, Z., Chen, K., Feng, Y., & Russell, A. G. (2019). Detailed Analysis of Estimated pH, Activity Coefficients, and Ion Concentrations between the Three Aerosol Thermodynamic Models. *Environmental Science and Technology*, *53*(15), 8903–8913. https://doi.org/10.1021/acs.est.9b00181

l.214: How did you determine the oxidation state?

We did not determine the oxidation state. Instead, we have assumed that all metal ions are present in their thermodynamically stable oxidation state. This has now been specified at line 234 of the revised manuscript: "For all metal ions the thermodynamically stable oxidation state was considered.".

l.248: What do you mean by scarce vegetation sources in this Namib region? Please rephrase this.

In this hyperarid region, there is hardly any vegetation, therefore, no sources of nutrient coming from vegetation.
This has now been clarified at line 269 of the revised manuscript: "…scarce vegetation sources in this hyperarid Namib region where there is hardly any vegetation."

l.256: What are the marine phytoplanktonic emissions of NO? Do you mean photochemical production of nitrogen oxides? Please rephrase this.

In the text, we have added "by photooxidation". It now reads :"The elevated nitrate concentrations can be attributed to the atmospheric processes affecting the important marine phytoplanktonic emissions of reactive nitrogen species (organic nitrogen and/or NO) ending, by photooxidation, in particulate nitrate formation (Altieri et al., 2021).

l.257: Please explain why weaker correlation of nitrate with MSA than oxalate supports this hypothesis.

Oxalate and MSA are tracers of secondary chemistry. Therefore, if nitrate is correlated to them, that strengthens the secondary origin of nitrate.
To avoid any confusion, we have rephrased, the sentence now reads: "The correlation of the nitrate concentrations with markers of secondary chemistry, such as oxalate and MSA ($R^2$=0.89 and 0.53, respectively) in our coastal fog samples supports this hypothesis.

l.281: Please explain the conversion of the unit in the method.

The passage has been clarified (line 303 of the revised manuscript) and now reads : "For the comparison, fog water-dissolved metal concentrations (measured in µM) were transformed into air concentrations (in ng/m$^3$) using the liquid water content (g H$_2$O/m$^3$). This allows for comparison of fog concentrations with dissolved metal concentrations in TSP. The ratio of fog concentrations to TSP concentrations is shown in Table 3. "

l.286 and Table 3: It is not clear whether the metal solubility is enhanced or the metal concentration is larger. How do you explain the lower values of the ratio between the mass concentration of dissolved metals in fog samples and soluble metals in the corresponding TSP samples than the unity? Please show the metal solubility in aerosol and fog samples, separately. Please also compare the solubility with the previous study at the HBAO.

In table 3 we compare dissolved metals in TSP and fog by calculating a ratio of their concentration in fog and TSP (converted in ng/m$^3$ of air sampled). Therefore, values above unity indicate a higher dissolved concentration in ng/m$^3$ in fog compared to TSP. Metal solubility in TSP samples is discussed in the companion paper by Formenti et al., this issue while only the dissolved fraction was measured in the fog samples.

l.297: How can you explain the lowest Fe solubility differences observed in the H1001 sample collected during daytime? Is this photochemical reduction consistent with the results on l.387, "photochemical reduction to Fe(II) was not a significant process involved in the enhanced iron solubility in the fog samples"? Please compare Fe solubility in fog samples and in the corresponding TSP samples quantitatively.

Solubility data for TSP are discussed in the companion paper by Formenti et al., this issue. For fog samples, only the dissolved fraction of the metals was measured. A lower solubility difference between TSP and fog for sample H1001 could be attributed to a lower MSA concentration in the fog sample (about one order of magnitude lower), partly due to dilution (higher liquid water content), compared to the other fog samples which would make the photoreduction process suggested by Desboeufs et al. 2024, less important in this case.

l.335 and Figure S6: Figure S6 did not show Fe(III) being predominantly complexed with oxalate. Please indicate the organic ligands and sampling date to compare with Figure 2.

We have now amended the sentence into: "…being complexed also with organics, in particular oxalate". We have also added information on the organic ligand to the caption of Figure S6 as well as sampling dates. The caption now reads: "Figure S6. Speciation of soluble metals in ALW in TSP from Henties Bay (Namibia) obtained using the model Visual MinteQ for (a) Al(III), (b) Ca(II), (c) Cu(II), (d) Fe(II), (e) Fe(III), (f) Mg(II), (g) Mn(II), (h) Ni(II), and (i) Zn(II). Samples shown cover the period 3-12 September 2017 (same as in Figure 2). For all metal ions, the organic ligand was oxalate."

l.375: How can you explain the lowest Fe solubility differences observed in the H1001 sample, which showed larger amounts of iron-oxalates in fog than aerosol, as opposed to previous studies cited on l.298? Please show the results for the fog sample H1002, which indicated the largest Fe solubility differences.

The previous study by Desboeufs et al. only analysed PM10 samples but not fog samples. Unfortunately, we do not have XANES measurements for the sample H1002, however, for sample H1001 the concentration of MSA is about one order of magnitude lower than in other fog samples so this could explain the lower solubility of Fe in that fog sample.

l.411: Can you tell the differences in colloidal materials between insoluble hydroxides and humic-like substances?

Unfortunately, it is not possible to identify organic compounds, such as humic-like substances with EDS analysis.

---

## Author Response (AR2)

We thank the editor and the reviewers for their helpful comments. We have addressed all the remaining comments, which further improved our manuscript. Below is our point-by-point response with details of the changes made.

**Editorial comments:**

a) Regarding figure 1: please ensure that the colour schemes used in your maps and charts allow readers with colour vision deficiencies to correctly interpret your findings. Please check your figures using the Coblis – Color Blindness Simulator (https://www.color-blindness.com/coblis-color-blindness-simulator/) and revise the colour schemes accordingly with the next file upload request.

We have now changed the middle and bottom panels to make them readable to colourblind readers.

b) Tables 3 and 4 contain coloured cells or/and coloured values. Please note that this will not be possible in the final revised version of the paper due to HTML conversion of the paper. When revising the final version, you can use footnotes or italic/bold font.

We have now removed the colour shading and used bold characters for highlighting instead in both tables.

**Reviewer 2 Akinori Ito**

**General comments**

When you use the word of solubility or dissolved fraction, the reader assumes that you measured total dissolvable and water-soluble or dissolved concentrations and then calculated the fractional solubility or dissolved fraction, as you mentioned previous studies such as on l.62 and l.399. However, you measured only the dissolved concentrations in the fog samples. Then you used dissolved fraction for the ratio of the dissolved concentration in fog to TSP. Please do not use the same word with different meanings.

We have now replaced "dissolved fraction" with either "dissolved metals" or "concentration of dissolved metals" throughout the manuscript.

**Specific comments**

l.65: How did you consider photo-reduction processes of marine biogenic emissions, photo-reductive dissolution of ferrihydrite by methanesulphinic acid, Fe(III)-carboxylate complexes in this paper, as a complement to those findings? Please discuss this on l.410 and consider rephrasing this.

We have realised that our phrasing could be misleading. We haven't specifically looked into photo-reduction processes, so we have now removed: "as a complement to those findings" and instead the sentence starts with "In this paper, we present a comparison...".

l.150: Please rephrase fraction by metal, or explain the fraction to what.

We have now changed the sentence as follows "dissolved metals are defined operationally as passing through a 0.2 µm filter which may include nano-sized particles as reported in previous studies (Giorio et al., 2025; Shi et al., 2009). One difference between the concentration of dissolved metals in fog ...".

l.360: This may be one of the reasons of enhanced dissolved Fe concentrations in fog samples compared to the TSP samples. Please discuss hydroxide in the TSP samples on section 4.5.

We thank the reviewer for this suggestion. We have now modified the sentence accordingly at now line 366: "the real speciation picture may be shifted in favour of a larger proportion of hydroxide formation for some metal ions, which could limit their ability to dissolve in water." We have also added a comment to section 4.5, at line 398: "In the case of TSP, iron oxides/hydroxides contribute 39-52% and iron clays contribute 38-52% (Table 4), making the majority of Fe insoluble. In fog samples, the contribution of these two classes of minerals is more variable, with iron oxides/hydroxides contributing 27-56% and iron clays contributing 32-64% (Table 4)."

l.409: The reference does not show more contribution of smaller particles with a higher soluble iron content to fog droplets compared to larger particles with a lower soluble iron content. Please check and correct this.

We have now rephrased the sentence as follows: "Additionally, Marcotte et al., (2020) showed that iron oxide solubility in mineral dust decreases with increasing particle size. Therefore, as activation to droplet is a size-dependent process, there may be a fractionation of soluble iron whereby, for example, smaller particles with a higher soluble iron content may contribute more to fog droplets compared to larger particles with a lower soluble iron content."